# Essential and Toxic Mineral Content and Fatty Acid Profile of Colostrum in Dairy Sheep

**DOI:** 10.3390/ani12202730

**Published:** 2022-10-11

**Authors:** Maria Francesca Guiso, Gianni Battacone, Linda Canu, Mario Deroma, Ilaria Langasco, Gavino Sanna, Eleni Tsiplakou, Giuseppe Pulina, Anna Nudda

**Affiliations:** 1Department of Agricultural Science, University of Sassari, Via Enrico de Nicola 9, 07100 Sassari, Italy; 2Department of Chemical, Physical, Mathematical and Natural Sciences, University of Sassari, Via Vienna 2, 07100 Sassari, Italy; 3Department of Animal Science and Aquaculture, Agricultural University of Athens, Iera Odos 75, 11855 Athens, Greece

**Keywords:** colostrum, essential minerals, fatty acid, immunoglobulins, antioxidant capacity, dairy ewe

## Abstract

**Simple Summary:**

Colostrum is of interest to the scientific community because of its nutritional and therapeutic capabilities. The aims of this study were to characterize the macro and micro composition of colostrum from Sarda dairy sheep and to compare it with the composition of the mature milk of the same breed. The results of this survey showed a large variation in the immunoglobulin concentration in colostrum, which could affect the acquisition of passive immunity by lambs. The strong correlation between immunoglobulin G and the total protein content suggests that this can be used to estimate the immunoglobulin content in sheep colostrum. The concentration of essential minerals is higher in colostrum than in milk as a result of mineral salt block supplementation at the end of gestation. Colostrum has a significantly different fatty acid profile than milk, and this is due to the specific needs of newborn lambs.

**Abstract:**

Colostrum is a major source of immunity in ruminants. It allows the transfer of antibodies from the mother to the fetus, and it is the exclusive source of nutrients for the newborn. The objectives of this study were (i) to characterize the macro and the micro composition of colostrum; (ii) to analyze the antioxidant capacity, fatty acid profile, and essential and toxic mineral content of colostrum; and (iii) to compare FA profiles and the amount of trace elements between colostrum and mature milk. For these purposes, samples of colostrum and milk were collected from a representative sample of animals from eight sheep dairy farms in the north of Sardinia (Italy). Fat, proteins, and seven essential and toxic minerals were measured in all samples of colostrum and milk. Furthermore, the FA profile was also measured in both matrices, while total antioxidant capacity was measured only in colostrum samples. The average amounts of fat and protein (TP) concentration in colostrum were 7.8% and 16%, respectively. Additionally, an average amount of 40 ± 20 g dm^−3^ was found for immunoglobulin G (IgG). As regards the antioxidant capacity of colostrum, a large variation was observed between samples from different farms for test 2, 2′-azino-bis(3-ethylbenzthiazoline-6-sulfonic acid) (ABTS), which was 30 ± 10% (mean ± standard deviation). High levels of selenium (Se), zinc (Zn), and copper (Cu) were found in colostrum (200 µg kg^−1^, 25,000 µg kg^−1^, and 1200 µg kg^−1^, respectively). A strong positive correlation between TP and IgG was observed (r = 91%). In colostrum, the amount of IgG is positively correlated with Se and Zn, as they are essential minerals to the immune system. The FA profile demonstrated higher levels of medium and long chain fatty acids in colostrum than in mature milk, and this is mainly true for arachidonic acid (ARA), ecosapentaenoic acid (EPA), docosapentaenoic acid (DPA), and docosahexaenoic acid (DHA). This study provided new information on the quality of colostrum in Sarda dairy sheep and showed the different composition of fatty acids between colostrum and mature milk.

## 1. Introduction

Colostrum in sheep provides immunoglobulins (Ig) that give passive immunity to newborn lambs [1,2] because the syndesmochorial placenta prevents the transfer of antibodies from the mother sheep to the fetus. In sheep colostrum, the primary immunoglobulin is IgG (more than 90% of total Ig), while immunoglobulins A (IgA) and M (IgM) are also present but at a much lower level [3]. Colostrum is also the only source of nutrients for newborns, as it is rich in fats, carbohydrates, and proteins. In addition, its rather complex composition resulting in a vital source of micronutrients (vitamins and minerals), antimicrobial (e.g., lactoferrin and lysozymes) and growth factors. Essential colostrum minerals also play a critical role in the prevention of nutritional pathology in lactating lambs, such as myodegenerative disease, known as white muscular disease, due to selenium (Se) deficiency [4]. Beyond Se, other essential minerals, such as copper (Cu), manganese (Mn), and zinc (Zn), have important antioxidant functions that are crucial to protect the integrity of the cellular membrane from oxidative stress. On the contrary, toxic minerals, such as lead (Pb) and cadmium (Cd), may be found in colostrum as a result of environmental pollution or feed contaminants. 

Ruminant colostrum has recently aroused considerable interest because of its potential nutritional and therapeutic effects also in humans [5,6,7], especially against inflammatory intestinal diseases [6,8]. Beyond its traditional use (e.g., a farm colostrum bank for lambs that have not been fed by their mothers) to reduce the use of antibiotics in newborns [9]), sheep colostrum can be used in human nutrition to make drinks or dietary supplements high in immunoglobulin. Sheep colostrum has also been a food resource for Mediterranean peoples. For example, in some parts of Italy it has been used to prepare a fresh cream or a medium-hard cheese (Casada), which is a peculiar ricotta known to be a traditional agri-food product [10]. Currently, there is little data available on the nutritional components of sheep colostrum, such as the content of mineral trace elements and the nutritional quality of fats. An adequate supply of trace minerals to transitioning ewes could significantly improve the composition and biological value of colostrum, which could affect the health of newborn lambs. In addition, it is known that organic supplements of trace elements, such as Zn, Cu, Mn, Se, and Fe, are used in transitional periods to greatly improve immunity and reproduction in dairy animals. 

The objective of this study was to characterize, in a reliable sampling of animals and farms, the gross composition; the Ig content; the amounts of essential minerals, such as Se, Cu, Mn, and Zn, of toxic elements, such as Pb and Cd, of an allergenic, such as Ni, and of the FA profile and the antioxidant potential of colostrum in Sarda dairy sheep. In addition, a comparison was carried out for FA profile and the amounts of minerals between colostrum and mature milk.

## 2. Materials and Methods

### 2.1. Ethical Practices

The experiment was approved by the Ethics committee of the University of Sassari (Prot. n. 139652 03/11/2021 with the authorization of Ministero della Salute n 676/2021-PR based on art. 31 D.lgs. 26/2014).

### 2.2. Animals and Experimental Design

The survey was conducted on eight sheep dairy farms in the north of Sardinia, Italy. They represented, both in terms of dimension and breeding technique, the situation of the dairy sheep industry in the zones (i.e., Sardinia and Central Italy) where the most important Italian ewe cheese, Pecorino Romano PDO, is produced [11]. Twelve sheep per farm, homogeneous for age (three years), were randomly selected for a total of ninety-six animals. Each ewe was identified with a tag and kept within the flock. Lambing was concentrated in November. 

### 2.3. Milk Sampling

Milk samples were collected from six ewes by manual milking in the morning time after 4 weeks from parturition, approximately 3–4 days after weaning the lamb to reduce the effects of stress separation on milk composition. Each milk sample was divided into aliquots and stored to be analyzed. An aliquot was used to determine the total protein and fat content (Milkoscan 6000; Foss Electric, Hillerød, Denmark) and the minerals, according to the procedure described in Section 2.6. The second aliquot was frozen at −25 °C until determination of the FA profile, which has been performed according to the method described in Section 2.5. 

### 2.4. Colostrum Sampling

A colostrum sample was collected from each ewe by manual milking within 24 h after lambing. Each colostrum sample was divided into aliquots and stored for the analysis.

### 2.5. Determination of IgG, Protein, Fat Content, and Fatty Acid Profile

The IgG content was determined at the laboratories of the “Bruno Ubertini” Istituto Zooprofilattico Sperimentale of Lombardia and Emilia Romagna, after caseins precipitations by means of electrophoresis followed by an UV-Vis quantification according to the literature methods [12,13]. Electrophoretic separation has been accomplished using Sebia Hydrasis LC and the kit Hydrasis Hydragel Protein (Sebia, Issy Les Moulineaux, France), whereas the quantification of the analyte has been performed by means of an autoanalyzer ILab 650 and the kit Total Protein (both from Instrumentation Laboratory Company, Lexington, MA, USA). 

The total nitrogen (TN) was measured using Kjeldhal method [14], and the total protein (TP) was calculated as TN × 6.38. Fat content was determined according to the Rose-Gottlieb method [15]. 

Total antioxidant capacity was measured by means UV-vis spectrophotometry with the ferric ion reducing antioxidant power (FRAP) and the 2, 2′-azino-bis (3-ethylbenzthiazoline-6-sulfonic acid) (ABTS) radical scavenging as described by Tsiplakou et al. [16].

The fatty acid profile was determined in 12 colostrum and 6 milk individual samples for each farm by gas-chromatography as detailed by Nudda et al. [17]. A gas-chromatograph Agilent model 7890 (Agilent Technologies, Santa Clara, CA, USA) equipped with a 7693 Autosampler (Agilent Technologies, Santa Clara, CA, USA) and a flame ionization detector (FID) operating to 225 °C was used. The stationary phase was a CP-Sil 88 capillary column (100 m × 0.250 μm i.d., 0.25 μm film thickness, Agilent Technologies, Santa Clara, CA, USA). Briefly, to 1 g of sample was added 0.4 cm^3^ of 25% of ammonia, 5 cm^3^ of hexane, and 1 cm^3^ of ethyl alcohol. The mixture was first vortexed for 2 min, and then centrifuged at 3000 rpm for 1 min, and the organic upper layer was separated by the aqueous one. The whole extraction procedure was repeated for a second time using 5 cm^3^ of hexane and 1 cm^3^ of ethyl alcohol 95% and again for a third by using only 5 cm^3^ of hexane. The organic layers were combined and heated at 40 °C under reduced pressure until solvent evaporation. The fatty acid methyl esters (FAME) were prepared by a base-catalyzed transesterification with the FIL-IDF 1999 standard procedure [18]. Briefly, about 25 mg of the extracted lipid was mixed with 1 cm^3^ of hexane (containing 0.5 mg of internal standard) and 1 cm^3^ of 2 mol dm^−3^ methanolic solution of KOH. The solution was vortexed for 2 min, then centrifuged for 1 min at 3000 rpm, and lastly added to 0.08 g of sodium hydrogen sulfate monohydrate. The supernatant was submitted again to centrifugation at 3000 rpm for 3 min and then injected in GC. Individual FAMEs were identified by comparison of the retention time of each analyte with that of a pure standard, according to Nudda et al. [17]. The nonadecanoic acid (C19:0) methyl ester Sigma Chemical Co., St. Louis, MO, USA) was used as internal standard for FAME quantification. 

The concentration of each fatty acid was expressed as per cent (%) of the total FAs amount, and the groups of FA were also calculated. The nutritional properties were valuated as the ratio between n-6 and n-3 and three indices; the atherogenic index (AI) and thrombogenic index (TI) were calculated as reported [19]: AI = [12:0 + (4 × 14:0) + 16:0]/[(PUFA) + (MUFA)]; TI = (14:0 + 16:0)/[(0.5 × MUFA) + (0.5 × n-6) + (3 × n-3) + (n-3: n-6)]; the hypocholesterolemic to hypercholesterolemic ratio (h:H) was calculated as [(sum of 18:1cis-9, 18:1cis-11, 18:2 n-6, 18:3 n-6,18:3 n-3, 20:3 n-6, 20:4 n-6, 20:5 n-3, 22:4 n-6, 22:5 n-3 and 22:6 n-3)/(14:0 + 16:0)].

The Δ9-desaturase indices (DI) were calculated according to Schennink et al. [20] to evaluate the effect of the different diets on the capacity of desaturating SFA to Δ9- UFA: C10 index = [C10:1/(C10:0 + C10:1)] × 100; C14 index = [C14:1 cis-9/(C14:0 + C14:1 cis-9)] × 100; C16 index = [C16:1 cis-9/(C16:0 + C16:1 cis-9)] × 100; C18 index = [C18:1 cis-9/(C18:0 + C18:1 cis-9)] × 100; CLA index = [CLA cis-9,trans-11/(C18:1 trans-11 + CLA cis-9,trans-11)] × 100; total index = [(C10:1 + C14:1 cis-9 + C16:1 cis-9 + C18:1 cis-9 + CLA cis-9,trans-11)/(C10:0 + C14:0 + C16:0 + C18:0 + C18:1 trans-11 + C10:1 + C14:1 cis-9 + C16:1 cis-9 + C18:1 cis-9 + CLA cis-9,trans-11)] × 100.

### 2.6. Determination of Minerals in Colostrum and Milk

The determination of the total amount of Cd, Cu, Mn, Se, Ni, Pb, and Zn in sheep’s samples of colostrum and milk were accomplished by means of a validated ICP-MS method. Samples were mineralized by means a Milestone Ethos Easy Labstation microwave oven (Milestone, Sorisole, Italy) and mineralized solutions were analyzed by means an inductively coupled plasma mass spectrometry (ICP-MS) spectrometer model NexION 300X equipped with an autosampler model S10 (Perkin Elmer, Monza, Italy), running under the Windows 7 operating system. Ultrapure (Type 1) water (specific resistance ≥ 18 MΩ) was used throughout the analytical procedure. The elemental standard solutions were by Carlo Erba (Milan, Italy) for Cd, Cu, Mn, Pb, Ni, Se, and Zn (100 mg dm^−3^ in 2% aqueous HNO_3_). The 67% aqueous solution of HNO_3_ and the 30% aqueous solution of H_2_O_2_ were both Ultrapure Normatom reagents (VWR International, Milan, Italy). The ERM-BD151 (skimmed milk powder) and the IAEA-A-13 (animal blood) Certified Reference Materials were by Merck, Milan, Italy and by IAEA, Vienna, Austria, respectively. The NexION ICP-MS tuning solution (2% HNO_3_ solution in water containing 1 µg dm^−3^ each of Be, Ce, Fe, In, Li, Mg, Pb, and U, code N8145051) and the NexION ICP-MS KED tuning solution (1% HCI solution in water containing Co, 10 µg dm^−3^ and Ce, 1 µg dm^−3^, code N8145052) were both purchased from Perkin Elmer Italia (Monza, Italy). The mineralization procedure (Appendix A), the instrumental settings (Appendix A) and the validation parameters (Appendix A) have been reported in the Appendix A.

### 2.7. Statistical Analysis

Differences in the concentrations of the colostrum components, IgG, fatty acid profile, and minerals among the farms were analyzed with one-way ANOVA and compared by Tukey test (SAS^®^ software). The relationship among minerals, total protein, fatty acids, and immunoglobulin were also evaluated. The statistical model to compare colostrum with mature milk for the FA profile and minerals included the fixed effects of type of secretion and farm, and the day of partum as a random factor.

## 3. Results and Discussion

The TP content in the colostrum (from 7.1% to 29%, mean 16%) was not significantly different among farms (*p* = 0.511), and it was markedly higher than that of milk (i.e., the 5.5%; *p* < 0.001), due to the presence of IgG (from 7.9 g dm^−3^ to 105 g dm^−3^, mean 40 g dm^−3^). Additionally, the concentration of IgG does not differ among farms (*p* = 0.379). The amounts of IgG measured here are consistent with those previously measured or the same breed (unpublished results) and are higher than that reported in others dairy sheep as Lacaune and Friesian, 28.9 and 28.8 g dm^−^^3^ [21,22]. Some research evidenced that lambs should intake an average of about 30 g of IgG in the first 24 h from birth to acquire the proper passive immunity [22,23,24]. Hence, considering an average concentration of 40 g of IgG per dm^−3^ of colostrum (Table 1), Sarda lambs could reach the immunity requirements suckling at least 0.75 dm^−3^ of colostrum. 

Colostrum’s antioxidant capacity, as measured by ABTS, showed high variability among farms. This is likely related to the different feeding techniques, which may have determined the passage of specific antioxidant substances, as phenolic compounds, from feeds to colostrum. On the contrary, the FRAP assay was not different among farms. The different results are likely because antioxidant assays can target a specific compound (as FRAP) or the total antioxidant capacity (as ABTS) given by the combined antioxidant capacities of all substances in a sample.

The contents of essential and toxic minerals in colostrum are reported in Table 2, whereas the correlation matrix of colostrum components is reported in Table 3. A wide variation of mineral level was found in colostrum collected from different farms. The concentration of Se in sheep colostrum (from 100 µg kg^−1^ to 350 µg kg^−1^, mean 200 µg kg^−1^) is greater than that found in colostrum of dairy cows [25] and goats [26] and markedly higher than values observed in the milk of the same breed (76.1 ± 40.6 µg kg^−1^ [27]) and other sheep breeds (28.4 ± 1.0 µg kg^−1^ [28]). Selenium deficiency and the consequent risk of white muscle disease (WMD) can be corrected by parenteral dosage of 0.1 mg of Se kg^−1^ body weight (BW) or by oral supplements, ensuring a concentration of 0.3 µg kg^−1^ [4]. This recommended Se level for lambs could be reached in our study by suckling 1.5 kg day^−1^ of colostrum containing 200 µg kg^−1^ of Se. On the contrary, the dosage of 0.4 µg kg^−1^ of Se can be too high and cause acute symptomatology characterized by sialorrhea, prostration, and dyspnea [29]. It is reported that Se has an important role in regulating the immunoglobulin and antioxidant capacity of colostrum [30,31]. This was also confirmed in this study where a significant positive correlation of Se and IgG has been observed (Table 3). 

Additionally, the Zn content in sheep colostrum (from 5000 µg kg^−1^ to 57,000 µg kg^−1^, mean 25,000 µg kg^−1^) is higher than reported in colostrum of goats [26] and of humans [32] and is markedly higher than reported in sheep milk [9,33]. Because of the role of Zn in immune function and in the teat keratin synthesis, it could reduce the susceptibility of the mammary gland to mastitis, which frequently occurs after the parturition [9]. However, the correct concentration of Zn in colostrum and milk is crucial for lambs’ growth as weight gain is dramatically reduced in lambs when Zn supplements provided only 0.05 mg Zn kg^1^ BW day^−1^, whereas an amount of 0.2 mg Zn kg^1^ BW day^−1^ ensures a good rate of growth [34]. 

The Cu content in sheep colostrum (from 130 µg kg^−^^1^ to 2800 µg kg^−1^, mean 1200 µg kg^−1^) is almost three times higher than the Cu content observed in goat colostrum [26] and in sheep milk (410 µg kg^−1^ [33]). Cu levels in sheep milk below 10 µg kg^−1^ favor the occurrence of swayback disease in newborn lambs [35], but values in colostrum measured in this study are more than 100-fold higher than this critical limit. Similar to Se, Mn, Zn, and Cu also have been reported to improve the levels of immunoglobulins and the antioxidant status of dairy animals [36]. However, in our study IgG correlates positively only with Zn, whereas a weak negative correlation has been observed with Cu, even though the increase in Ig in the blood of lambs supplemented with Cu has been reported [37]. 

The mean concentrations of heavy metals, such as Pb and Cd, in colostrum are markedly lower than the maximum limits indicated by the European Union for raw drinking milk and baby foods (i.e., 20 µg kg^−1^ and 10 µg kg^−1^, respectively) [38]. Sheep’s intake of toxic metals can result from contaminated feed or soil where the animals graze, resulting in the exposure of suckling lambs. 

The concentration of nickel in sheep colostrum was below the value reported in milk [39,40] and below the permissible limit (0.43 mg dm^−3^) set by the World Health Organization (WHO), Geneva, Switzerland. 

According to the correlation matrix (Table 3), there is clearly a strong positive correlation between IgG and TP. Therefore, TP may be used as a reliable parameter to estimate the IgG content of colostrum samples taken on the day of parturition. Moreover, IgG has significant correlations with Se and Zn. This correlation is based on the fact that the two elements are essential to the immune system [41]. Unexpectedly, IgG also shows a slightly positive correlation with Ni. Conversely, there is a weak negative correlation between TP and fat, probably due the low synthesis of components in the mammary gland compared to the high passage from blood, mostly IgG, through the paracellular route because the leakage of tight junction of mammary epithelium cells. A weak negative correlation between IgG and Cu has been also observed (*p* < 0.05). Immunosuppression due to the excessive Cu intake has been previously reported in sheep [40]. Another small negative correlation was observed between IgG and Cu (*p* < 0.05), confirming the literature studies where immunosuppression due to excessive Cu intake was observed [42].

A positive and significant correlation of FRAP with protein and Ig contents has been found. A positive and significant correlation between ABTS and fat was observed as well. This is consistent with findings from the literature showing that whole cow milk showed a higher level of ABTS inhibition than skimmed milk [43,44]. It is interesting to note that there is no correlation between FRAP and ABTS (Table 3). This finding is not surprising because the chemical composition of colostrum can affect the sensitivity of these methods. Indeed, it has been reported that the ABTS method can better determine the total oxidant status of whey and whole milk and is more sensitive to casein, whereas the FRAP method is better addressed for measuring the antioxidant capacity of serum protein [44]. In addition, discrepancies in the results obtained from the same material analyzed using different methods have been evidenced in the literature as a function of the nature of the different antioxidant groups [45].

The comparison between the amounts of minerals found in colostrum and milk is reported in Figure 1. While the concentration of Ni and Pb did not differ in milk and in colostrum, those of Cd, Se, Cu, and Mn were significantly higher in colostrum rather than in milk. In particular, the concentration of Zn in colostrum is three times that of milk. This might be related to the mineral salt blocks supplementation, containing mainly Zn oxide, Mn oxide, and Na selenite, that are usually made available to sheep in the last two months of gestation, to improve the overall performance of the mothers and offspring. The Cu is not present in mineral blocks destined for ewes. These ruminants are highly sensitive to Cu toxicity due to their low physiological needs and their poor ability to manage excessive intakes of this element. In fact, sheep easily accumulate Cu in the liver, which could be suddenly mobilized in the circulating blood, causing severe symptoms of toxicity. Recommended dietary intakes of Cu in sheep range from 3000 to 10,000 µg kg^−1^, whereas amounts greater than 15,000 µg kg^−1^ are toxic [46]. The analysis of Cu in the diet showed a concentration of 6000 ± 3000 µg kg^−1^ (mean ± sd) and, therefore, within the recommended levels for safety. 

The concentration of Cu observed in colostrum is seven times greater than in milk (*p* < 0.001). This could likely be due to the dietary inclusion of Cu-rich ingredients in late gestation animals as soybean meal, which contains significant concentrations of this mineral. 

The fatty acid profile of colostrum compared to that of milk are reported in Table 4, whereas Appendix A in the Appendix A shows the average concentration of all FAs determined in the colostrum of each farm. The data presented in Appendix A show high variability among farms for almost all FA, with the exception of some very low-level fatty acids in colostrum, e.g., the isoC13:0 and the C8:1 t10.

In colostrum, the C16:0, the C18:1c9, the C14:0, and the C18:0 are the most abundant FAs. In this matrix, medium-chain fatty acids (MCFA) and long chain fatty acids (LCFA) dominate on the short-chain FAs (SCFA), which are less abundant in colostrum rather than in mature milk. This is coherent to what is observed in the literature for the comparison between bovine colostrum and mature milk [47]. Furthermore, the concentration of odd chain FAs (OCFA) and branched-chain FAs (BCFA) is lower in colostrum rather than in milk, and the same is observed for the content of C18:1 t11 and conjugated linoleic acid, CLAc9t11. The values observed in milk were in line with previous reports in Sarda dairy sheep [48,49]. In this case, such behavior is not in agreement with that observed in dairy cows, in which no differences (e.g., for C18:1 t11) or no decreases in concentration (e.g., for C17:0 and for CLAc9t11) passing from milk to colostrum has been observed [47]. 

In the long chain polyunsaturated FAs (PUFA), the arachidonic acid (ARA), the eicosapentaenoic acid (EPA), the docosapentanoic acid (DPA), and the docosahexaenoic acid (DHA) were most abundant in colostrum rather than in mature milk, and this is likely due to the specific needs for newborn lambs. In fact, ARA and DHA are essential and structural constituents of cellular membranes and are mainly required for the growth and function of the brain and nervous system [50,51]. The ARA and DHA contents in colostrum were two times higher than that in the mature milk. This fact is important, since the newborn has very low ability to elongate linoleic acid (LA) to ARA and alpha-linolenic acid (ALA) to DHA; therefore, the high amounts of LC-PUFA in colostrum could be of crucial advantage to lambs. Finally, the concentrations of odd- and branched-chain FAs (OBCFA) were lower in colostrum rather than in milk. This is probably due to the reduction in dietary intake in late gestation and the beginning of lactation, which may result in a reduced microbial activity of the rumen. 

All desaturase indexes were higher in colostrum than milk; different minerals were reported to affect D9-desaturase activity [51,52,53,54], and therefore, an increase in the expression of desaturase due to the direct or indirect effects of minerals could be a possible key of explanation. The atherogenic index (AI) for lipids and the h:H, widely reported in the literature as a dietary risk indicator for cardiovascular disease, were slightly but significantly lower in colostrum than milk due to the higher content of LCFA and MUFA. For this last class of FAs, the C18:1c9 shows the highest difference in concentration between colostrum and mature milk.

The comparison of the FA acid profile of colostrum and the mature milk allows to achieve information on the specific metabolism of ewes and udder during the transition period. In particular, the higher content of FAs derived from body fat, the lower amount of OBCFA due to reduced rumen activity in the peripartum period, and the higher D9-desaturase activity provided evidence of the metabolism of animals after parturition. 

## 4. Conclusions

For the first time the composition of colostrum of Sarda dairy ewes has been characterized in terms of TP, fat, IgG concentration, amount of seven minerals (i.e., Cd, Cu, Mn, Ni, Pb, Se, and Zn), the FA profile, and of the antioxidant properties, measured in terms of both FRAP and ABTS. The strong correlation between IgG and TP shows that the latter parameter can be used to estimate the IgG content of colostrum in sheep farming. IgG is also positively correlated with Se and Zn, two “well-known” stimulators within the animal’s immune system. The average amounts of toxic elements were always less than the EU limits posed for these elements in milk. The FA profile of colostrum shows high variability between farms and is significantly different from that measured in mature milk. Trace minerals may serve as an important marker of the nutritional and healthy quality of colostrum. From a practical point of view, some considerations can be drawn from the results of this study. First, more attention should be exercised in the amount of some ingredients included in the diet of late pregnant sheep that could be rich in Cu (e.g., soybean meal), in order to avoid an excess of Cu in the colostrum. In addition, monitoring the concentration of IgG in bulk colostrum can result in inadequate immunity acquired by some lactating lambs because of the large variability observed in the concentration of Ig in colostrum. Finally, the lack of correlation between markers of oxidation in colostrum suggests that the use of different analytical methods is necessary to describe its antioxidant capability.

## Figures and Tables

**Figure 1 animals-12-02730-f001:**
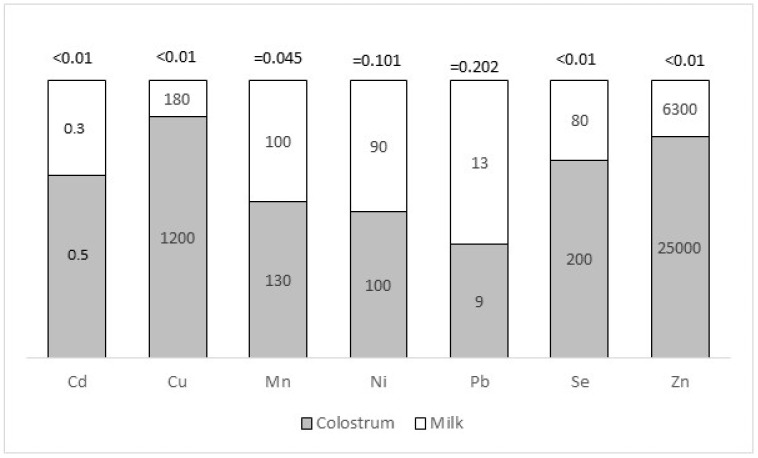
Concentration (µg kg^−1^) of essential and toxic minerals in colostrum (dark bars) and mature milk (light bars) sampled in eight farms. The concentration of minerals is reported into the bars. The statistical differences are reported as the *p* value in the top of the bars.

**Table 1 animals-12-02730-t001:** Total protein, fat, IgG content, and antioxidant properties of colostrum sampled in eight farms.

			Colostrum Composition		
Farm	TP (%)	IgG (g dm^−^^3^)	Fat (%)	FRAP (µmol Ascorbic Acid dm^−3^)	ABTS (% Inhibition)
A	17	40	5.3 ^c^	3.5	20 b
B	16	40	7.6 ^abc^	3.4	20 b
C	14	30	7.7 ^abc^	nd	nd
D	15	40	9.6 ^a^	2.3	20 b
E	15	40	6.9 ^abc^	2.4	20 b
F	19	50	5.5 ^bc^	3.1	20 b
G	14.5	40	9.1 ^ab^	2.2	40 a
H	16	40	10 ^a^	3.4	50 a
Mean of all samples	16	40	7.8	2.9	30
SD	5	20	3.2	1.4	10
Min	7.1	7.9	2.5	0.7	10
Max	29	105	18	8.5	58
Pvalue	0.511	0.379	<0.001	0.120	<0.001
SEM	0.54	2.20	0.36	0.15	1.53

Means in the same row with different superscripts differ (*p* < 0.05). Average amounts reported are rounded according to the number of significant digits of the relevant standard deviation, while statistical tests have been accomplished on unrounded data. A–H = the farms used in the survey; TP = total protein (TN × 6.38); IgG = immunoglobulin G; FRAP = ferric reducing antioxidant power; ABTS = 2, 2′-azino-bis(3-ethylbenzthiazoline-6-sulfonic acid); nd = not determined.

**Table 2 animals-12-02730-t002:** Concentration (µg kg^−^^1^) of essential and toxic minerals in colostrum sampled in eight farms.

			Minerals				
Farm	Cd	Cu	Mn	Ni	Pb	Se	Zn
A	0.4 ^ab^	660 ^d^	140 ^b^	90 ^ab^	10	300 ^ab^	26,000
B	0.8 ^a^	1200 ^abc^	70 ^c^	110 ^ab^	12	200 ^ab^	32,000
C	0.5 ^ab^	1600 ^a^	240 ^a^	80 ^b^	7	200 ^ab^	23,000
D	0.6 ^ab^	880 ^bcd^	250 ^a^	120 ^a^	11	300 ^ab^	23,000
E	0.4 ^ab^	1400 ^ab^	74 ^bc^	80 ^b^	7	200 ^ab^	18,000
F	0.6 ^ab^	670 ^cd^	74 ^bc^	80 ^b^	7	100 ^b^	27,000
G	0.6 ^ab^	1300 ^abc^	100 ^bc^	110 ^ab^	9	350 ^a^	25,000
H	0.3 ^b^	1600 ^a^	60 ^c^	100 ^ab^	11	200 ^ab^	22,000
Mean	0.5	1200	130	100	9	200	25,000
SD	0.3	600	90	30	5	200	11,000
Min	0.1	130	33	39	3	20	5000
Max	1.5	2800	400	190	32	1000	57,000
*p*-value	0.0083	<0.0001	<0.0001	0.0016	0.0417	0.0282	0.217

Means in the same row with different superscripts differ (*p* < 0.05). Average amounts reported are rounded according to the number of significant digits of the relevant standard deviation, while statistical tests have been accomplished on unrounded data. Cd = cadmium; Cu = copper; Mn = manganese; Ni = nickel; Pb = lead; Se = selenium; Zn = zinc.

**Table 3 animals-12-02730-t003:** Correlation matrix between trace minerals and immunoglobulins, protein and fat content, and antioxidant power of colostrum.

	Se	Cu	Mn	Zn	Ni	Pb	Cd	TP	IgG	Fat	FRAP
**Cu**	0.053	1									
**Mn**	0.220	−0.124	1								
**Zn**	0.469 **	−0.115	0.114	1							
**Ni**	0.428 **	−0.087	0.336 **	0.486 **	1						
**Pb**	0.127	−0.020	−0.003	0.135	0.433 **	1					
**Cd**	0.041	0.017	−0.035	0.154	0.263 *	0.059	1				
**NT**	0.492 **	−0.208	0.113	0.680 **	0.359 **	0.039	0.051	1			
**IgG**	0.447 **	−0.254 *	−0.005	0.555 **	0.250 *	0.015	0.107	0.912 **	1		
**Fat**	−0.177	0.247 *	0.029	−0.063	0.058	0.014	−0.021	−0.331 *	0.412 **	1	
**FRAP**	0.135	−0.111	−0.038	0.310 **	−0.028	−0.126	−0.058	0.285 *	0.256 *	−0.035	1
**ABTS**	0.051	0.39 **	−0.219	−0.072	−0.048	0.087	−0.099	−0.063	−0.182	0.44 **	−0.025

Correlation significantly different from zero: ** *p* < 0.001; * *p* < 0.05.

**Table 4 animals-12-02730-t004:** Fat content and fatty acid profile of colostrum (% on the total of FAs) and mature milk sampled in eight farms.

	Type			*p*-Value	
	Colostrum	Milk	SEM	Type	Farm
Fat, %	8	5.8	0.4922	0.0081	0.0021
FA (% on Total FAs)					
Short chain FA					
C4:0	1.8	2.4	0.0656	<0.0001	0.0017
C6:0	0.8	2.0	0.0417	<0.0001	0.0006
C7:0	0.05	0.08	0.0048	0.0005	<0.0001
C8:0	0.6	2.1	0.0464	<0.0001	0.0063
C9:0	0.10	0.15	0.0095	0.0441	<0.0001
C10:0	1.8	7	0.1744	<0.0001	0.0031
C10:1	0.06	0.09	0.0052	0.0001	0.0226
Medium chain FA					
C11:0	0.12	0.32	0.0126	<0.0001	0.0041
C12:0	2.0	4.2	0.1112	<0.0001	0.0097
isoC13:0	0.01	0.02	0.0008	<0.0001	0.0441
anteisoC13:0	0.03	0.04	0.0023	0.0064	0.0009
isoC14:0	0.06	0.12	0.0039	<0.0001	0.3183
C14:0	11	11	0.4014	0.9561	0.0001
isoC15:0	0.18	0.31	0.0071	<0.0001	0.0398
anteisoC15:0	0.20	0.54	0.0116	<0.0001	0.1539
C14:1c9	0.4	0.17	0.0353	<0.0001	<0.0001
C15:0	0.6	1.1	0.0251	<0.0001	0.0002
C15:1	0.03	0.08	0.003	<0.0001	<0.0001
isoC16:0	0.20	0.35	0.0083	<0.0001	0.1687
C16:0	29	24	0.8182	<0.0001	<0.0001
isoC17:0	0.42	0.51	0.0132	<0.0001	<0.0001
anteisoC17:0	0.47	0.48	0.0155	0.7088	<0.0001
C16:1c9	1.6	0.8	0.1189	<0.0001	0.0001
C17:0	0.8	0.72	0.0264	0.0002	0.0004
isoC18:0	0.12	0.07	0.005	<0.0001	0.0012
C17:1c9	0.4	0.19	0.0148	<0.0001	0.0006
Long chain FA					
C18:0	7	10	0.3547	<0.0001	0.0162
C18:1t4-8	0.21	0.28	0.0082	<0.0001	0.0037
C18:1t9	0.21	0.25	0.009	0.0004	<0.0001
C18:1t10	0.3	0.4	0.0273	<0.0001	0.0367
C18:1t11	0.7	1.6	0.0752	<0.0001	0.0059
C18:1t12	0.23	0.4	0.0162	<0.0001	0.008
C18:1t13:t14	0.3	1.1	0.0391	<0.0001	0.0306
C18:1c9	30	17	1.0669	<0.0001	<0.0001
C18:2n6 (LA)	2.5	2.1	0.1102	0.0001	<0.0001
C20:0	0.22	0.26	0.0078	<0.0001	<0.0001
C18:3n6	0.05	0.05	0.0037	0.0631	0.0044
C20:1c9	0.03	0.02	0.0012	<0.0001	0.0002
C18:3n3 (ALA)	0.4	0.7	0.0349	<0.0001	<0.0001
CLAc9t11	0.7	0.8	0.0422	0.0043	<0.0001
CLAt10c12	0.03	0.05	0.0031	<0.0001	0.0003
CLAt12t14	0.01	0.04	0.0021	<0.0001	0.0065
CLAt11t13	0.03	0.06	0.0034	<0.0001	0.042
CLAt9t11	0.02	0.03	0.002	<0.0001	0.0098
C18:4n3	0.01	0.01	0.0005	0.0002	0.0229
C20:2n6	0.03	0.02	0.0011	<0.0001	0.3811
C20:3n9	0.07	0.10	0.0051	<0.0001	0.4713
C22:0	0.07	0.15	0.0048	<0.0001	0.0035
C20:3n6	0.04	0.03	0.0014	<0.0001	0.0006
C20:4n6 (ARA)	0.31	0.14	0.0116	<0.0001	<0.0001
C23:0	0.02	0.08	0.0025	<0.0001	0.0252
EPA	0.07	0.05	0.0033	<0.0001	<0.0001
DPA	0.17	0.09	0.0084	<0.0001	<0.0001
DHA	0.06	0.03	0.0034	<0.0001	<0.0001
Groups of FA					
SCFA	5	14	0.2799	<0.0001	0.0015
MCFA	48	46	1.3171	0.25	<0.0001
LCFA	47	40	1.4586	0.0003	<0.0001
SFA	58	69	1.1462	<0.0001	<0.0001
MUFA	37	25	1.0343	<0.0001	<0.0001
PUFA	6	6.1	0.1953	0.152	0.0103
UFA	42	31	1.1463	<0.0001	<0.0001
OCFA	1.7	2.4	0.0455	<0.0001	0.0014
BCFA	1.7	2.4	0.0458	<0.0001	0.0034
OBCFA	3.4	4.9	0.0769	<0.0001	0.0023
PUFA6	3.0	2.5	0.1208	<0.0001	<0.0001
PUFA3	0.8	0.9	0.0464	0.0078	<0.0001
n6/n3	4	3	0.3021	<0.0001	<0.0001
n3/n6	0.27	0.4	0.0213	<0.0001	<0.0001
CLA	0.8	1.1	0.0479	<0.0001	<0.0001
TFA	3.4	6	0.2146	<0.0001	0.0012
TFA (without VA)	2.7	5	0.1519	<0.0001	0.0013
Indexes					
AI	1.9	2.3	0.1329	0.0071	<0.0001
TI	1.9	2.1	0.1237	0.1187	<0.0001
h/H	0.9	0.60	0.049	<0.0001	<0.0001
DI C10:1	4	1.3	0.2513	<0.0001	<0.0001
DI C14:1	3	1.5	0.178	<0.0001	0.0018
DI C16:1	5	2.9	0.2219	<0.0001	0.0081
DI C18:1	80	63	0.7329	<0.0001	0.1824
DI CLA	51	35	0.9838	<0.0001	0.5566

Average amounts reported are rounded according to the number of significant digits of the relevant standard deviation. SD = standard deviation; ΣFAs = sim of all FAs; FAME = fatty acid methyl ester; SA = stearic acid; LA = linoleic acid; ALA = linolenic acid; ARA = arachidonic acid; EPA = eicosapentaenoic acid; DPA = docosapentaenoic acid; DHA = docosahexaenoic acid; SFA = sum of the individual saturated fatty acids; UFA = sum of the individual unsaturated fatty acids; MUFA = sum of the individual monounsaturated fatty acids; PUFA = sum of the individual polyunsaturated fatty acids; OCFA = odd-chain fatty acids; BCFA = branched-chain fatty acids, sum of iso- and anteiso-FA; OBCFA = odd- and branched-chain fatty acids, sum of odd-, iso-, and anteiso-FA; SCFA, short-chain fatty acids (sum of individual fatty acids from C4:0 to C10:0); MCFA = medium-chain fatty acids, sum of the individual fatty acids from C11:0 to C17:0; LCFA = long-chain fatty acids, sum of the individual fatty acids from C18:0 to DHA; PUFA n-3 and PUFA n-6 = sum of individual n-3 and n-6 fatty acids, respectively; CLA = sum of individual conjugated linoleic acids; TI = thrombogenic index; AI = atherogenic index; h:H = hypocholesterolemic to hypercholesterolemic ratio.

## Data Availability

The data presented in this study are available on request from the corresponding author.

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
