# Peer review of "Essential and Toxic Mineral Content and Fatty Acid Profile of Colostrum in Dairy Sheep"

_animals, 2022, doi:10.3390/ani12202730_

Round 1
Reviewer 1 Report
This article was conducted to analyze the nutritional contents of the colostrum in dairy sheep and made a comparison of fatty acids between colostrum and mature milk. The research is very meaningful, and the experiment data is abundant. But I think the analysis of the experiment and the conclusion of the article is not very appropriate.
Here are the specific suggestions.
1. The nutritional indicators of colostrum are not complete. Why there is no lactose? Why immune index only has IgG?
2. Why only compare the fatty acid differences between colostrum and mature milk? In my opinion, it is better to compare all nutritional indexes between colostrum and mature milk.
3. Line 99, "The fatty acid profile was determined in 6 samples for each farm". Each farm has twelve animals, is there for fatty acid detection only use six animal samples? It is also not very clear how many samples to use for other detection indicators.
4. There is no mature milk sampling. We should know when and how you do this sampling.
5. The data in Table 4 seems too much, and it is out of focus. And the data didn't show the differences among the eight farms.
Author Response
Reviewer 1:
The nutritional indicators of colostrum are not complete. Why there is no lactose? Why immune index only has IgG?
AU: We thank so much the reviewer for these sharp observations. As a matter of fact, the concentration of the lactose in colostrum have been not measured. The sampling, entirely held in December 2020, coincided unfortunately with the Christmas lockdown period, and the planned determination of lactose have not accomplished due the lockdown of our labs. Hence, due to the non-repeatability of the sampling, we cannot provide in this paper these data. On the other hand, the determination of lactose has been performed as planned on the milk samples, which have been sampled in January 2021. As far as the immune index is concerned, literature (see ref. 3 in the revised version of the paper) reported that, in sheep colostrum, the IgG represents more than 90% of the total amount of Ig’s, as reported in lines 51-53 of the revised version of the paper.
- Why only compare the fatty acid differences between colostrum and mature milk? In my opinion, it is better to compare all nutritional indexes between colostrum and mature milk.
We agree. Data relative to the mean amount of total proteins has been added in lines 194-195 of the revised version of the paper, whereas the concentration of minerals in milk has been reported in the new figure, Figure 1. On the other hand, the amounts of fat-related nutritional indexes (i.e. the total fat and the FAs) have already present in Table 4.
- Line 99, "The fatty acid profile was determined in 6 samples for each farm". Each farm has twelve animals, is there for fatty acid detection only use six animal samples? It is also not very clear how many samples to use for other detection indicators.
and
- There is no mature milk sampling. We should know when and how you do this sampling.
AU: We thank so much the reviewer #1 to highlight a misleading part in the experimental section of our paper. The principal aim of this study was to characterize in a reliable way the colostrum from Sarda sheep’s. Hence, twelve animals were selected in each of the eight farms chosen for the experiment, in order to ensure the best representativity of data obtained. As a general rule, all animals were used to obtain data from colostrum samples. On the other hand, the composition of milk from Sarda sheep’s has been largely known as far as its macrocomposition and the FA profile are concerned. In this case we have chosen to limit the number of samples analyzed to six samples for each farm, as reported in the new paragraph (i.e. 2.3. Milk sampling, lines 99-107 on the revised paper). In this paragraph we have added also the conditions of milk sampling. It has been done at 4 weeks postpartum, approximately few days after the lamb was weaned, to reduce the effect of stress on milk composition, due to lamb separation.
- The data in Table 4 seems too much, and it is out of focus. And the data didn't show the differences among the eight farms
The aim of Table 4 was to highlight analogies and differences in the FA profile of both colostrum and milk. On the other hand, we realize that the version of this table reported in the submitted version of the paper is rather cumbersome, hence we acknowledge the suggestion of the Reviewer #1. A simplified version of Table 4 has been inserted in the revised version of the paper. In comparison to the previous version, 48 minority FAs were deleted, and the FAs have been grouped according to the length of the chain (i.e. short, medium and long -chain FAs). In addition, we have inserted in the Supplementary material a table (Table S4) reporting the average concentration of all FAs determined in colostrum of each farm to show differences among farms.
Reviewer 2 Report
Comments and Suggestions for Authors
I would like to thanks Authors for proposing a very interesting study. It is better for the authors to use more concise language. In the manuscript, some language editing is required to correct the grammar and syntax errors. Some expressions need to be more precise and concise to avoid confusing readers.
Simple Summary is laconic, the description could be more elaborate.
Abstract:
Line 19: “...the macro e micro composition…” change to “…the macro AND micro composition…”
Line 22-23: 10 samples, from 8 different herds, from the large region of North Sardinia - a large variety of samples was introduced in the experiment (not enough trials). Please justify such a choice of trials in the conducted experiment.
Line 26: “...an average of 40 ± 20 g dm-3 was found for Immunoglobulin G (IgG).” The value of SD is 50%. The number of colostrum samples should be increased or extreme results should be discarded.
Introduction:
Line 72: Why were only these two analyzes carried out on the milk samples? Why were the analysis of essential trace elements and toxic elements not performed on milk?
Materials and Methods:
Line 82-83: “Twelve animals per farm, homogenous for age (3 years) were randomly chosen for a total of 96 samples.”
In the abstract, the authors mention 10 animals, here 12 animals. Which value is correct? You mention 96 samples in total, add this information in the Abstract (this is response to my earlier suggestions for a small number of trials).
Line 98: Correct “…by Tsiplakou ET al. [14].”
Line 196: Explain the abbreviation - WMD
Line 214-223: It is worth mentioning about the toxicity of copper in sheep nutrition and its limit values.
Line 242-243: Please expand / explain this negative correlation.
Line 294: Explain the abbreviation - OBCFA
Line 297: Table 4 contains a lot of data (2 pages!). I suggest dividing table 4 into smaller tables, e.g. short-chain acids, medium-chain acids, etc ... This will make it easier for the reader to follow the results
The conclusion is correct, however in the presented manuscript I miss the experience aspect. Despite the interesting results this manuscript is only characterization of the colostrum composition of Sarda dairy ewes.
In its current form the manuscript should not be accepted for publication in the Animals journal.
Author Response
Reviewer 2:
Comments and Suggestions for Authors
I would like to thanks Authors for proposing a very interesting study. It is better for the authors to use more concise language. In the manuscript, some language editing is required to correct the grammar and syntax errors. Some expressions need to be more precise and concise to avoid confusing readers.
We agree. The paper was thoroughly checked by a mother-tongue expert reader in order to increase its readability.
Simple Summary is laconic, the description could be more elaborate.
We agree. The simple summary has been expanded according to the suggestion of the Reviewer #2 (see lines 16-21 of the revised paper)
Line 19: “...the macro e micro composition…” change to “…the macro AND micro composition…”
Many thanks for the reporting; it has been amended in the revised version of the manuscript
Line 22-23: Eighty colostrum samples and 10 samples, from 8 different herds, from the large region of North Sardinia - a large variety of samples was introduced in the experiment (not enough trials). Please justify such a choice of trials in the conducted experiment.
We apologize, but lines 22-23 of the submitted paper contained a typing error: the sheep chosen for each farm were twelve and not ten. The mistake has been corrected in the revised version of the paper. We have chosen to survey eight farms placed in a quite homogeneous area of Sardinia considering firstly the presence of previous recorded data of the flock and the availability of farmers to give us their data during the entire lactation and to permit periodic samples. The presence of selected animals within the flock and the correct identification for the entire duration of the project require efforts from farmers. that not everyone is willing to give. However, regarding the main objective of the paper (e.g. colostrum) the farms resulted perfectly randomized and representative of the average technical and environmental conditions in which Sardinian dairy sheep farms are managed (Pulina et al., 2018 – doi: 10.3168/jds.2017-14015).
Line 26: “...an average of 40 ± 20 g dm-3 was found for Immunoglobulin G (IgG).” The value of SD is 50%. The number of colostrum samples should be increased or extreme results should be discarded.
The high variability found in colostrum is a result of the work. Moreover, we do not have objective criterion for judging outliers. For these reasons, we preferred to keep explicit all the found variability.
Introduction:
Line 72: Why were only these two analyzes carried out on the milk samples? Why were the analysis of essential trace elements and toxic elements not performed on milk?
We agree. This sentence is quite misleading, and it has been completely rewritten. In addition, all the data solicited by the Reviewer #2 have been added in the revised version of the paper.
Materials and Methods:
Line 82-83: “Twelve animals per farm, homogenous for age (3 years) were randomly chosen for a total of 96 samples.” In the abstract, the authors mention 10 animals, here 12 animals. Which value is correct? You mention 96 samples in total, add this information in the Abstract (this is response to my earlier suggestions for a small number of trials).
We agree, and sorry for our mistake. The animals considered for each farm are twelve and not ten (see also reply to the remark relative to lines 22-23)
Line 98: Correct “…by Tsiplakou ET al. [14].”
and
Line 196: Explain the abbreviation – WMD
We agree. The corrections have been inserted in the revised version of the paper.
Line 214-223: It is worth mentioning about the toxicity of copper in sheep nutrition and its limit values.
We agree. The recommended intake of copper and the toxicity level have been added into the text (please see lines 305-312 of the revised version of the manuscript)
Line 294: Explain the abbreviation – OBCFA
Done. The abbreviation has been explained in the revised version of the paper
Line 297: Table 4 contains a lot of data (2 pages!). I suggest dividing table 4 into smaller tables, e.g. short-chain acids, medium-chain acids, etc ... This will make it easier for the reader to follow the results
We agree. Similar suggestion has been done from reviewer 1. The number of FA have been reduced in table 4 (we remove 48 FA). The FA are grouped to simplify the reading, but we prefer to maintain all FA in the same table.
The conclusion is correct, however in the presented manuscript I miss the experience aspect. Despite the interesting results this manuscript is only characterization of the colostrum composition of Sarda dairy ewes.
Thank for the useful suggestion. The conclusion has been revised and some practical suggestion have been added (please see lines 389-400 of the revised version of the paper).
Reviewer 3 Report
The Article “Essential and toxic minerals content and fatty acid profile of colostrum in dairy sheep” is an important contribution to scientific knowledge of the characteristics of colostrum in sheep, which initiates a very interesting line of study because it is applicable in the animal production sector.
The study is based on a good analytical basis, with little characterized elements in colostrum (minerals, antioxidants, etc.).
In general, this is a good study, which should be improved by expanding the discussion on the results achieved.
Especially, please consider the following considerations:
Check the spelling of the manuscript (parentheses, etc.).
33-34: Acronyms (ARA, EPA, DPA and DHA) must be accompanied by the full name. They are only described in Table 4.
This also happens with other acronyms throughout the text, such as for example SFA, UFA, MUFA, PUFA
91-92: The use of this method should be supported by a bibliographical reference.
99-122: Review the text of the extraction procedure of fames; is it repeated two or three times. It is not clear in the text.
173-175: Discuss and better explain the following sentence: Applying this principle, the lambs of this study should ingest 0.75 dm-3 of colostrum containing 40 g of IgG dm-3 to meet the immunity requirements.
178-181: Review the explanation of FRAP/ABTS, based on inconclusive FRAP results.
190-222: It is recommended to provide some explanations about the upper levels of Se, Zn and Cu of this study. Can they be due to fed, analysis, etc?
269: Please check the format.
247-248: Are these results or is it a reference?: More specifically, skimmed milk has a 6 % (ABTS) lower total antioxidant capacity than whole milk.
273-295: The results of the fatty acid index should be discussed (TI, AI, etc).
323-324: I think the conclusion is very pretentious that the amounts of trace elements could be an important marker of environmental contamination of an area in which animals are breed. Further analysis (soil, water, etc.) would be required to reach this conclusion.
Author Response
Reviewer 3:
The Article “Essential and toxic minerals content and fatty acid profile of colostrum in dairy sheep” is an important contribution to scientific knowledge of the characteristics of colostrum in sheep, which initiates a very interesting line of study because it is applicable in the animal production sector.
The study is based on a good analytical basis, with little characterized elements in colostrum (minerals, antioxidants, etc.).
In general, this is a good study, which should be improved by expanding the discussion on the results achieved.
Especially, please consider the following considerations:
Check the spelling of the manuscript (parentheses, etc.).
and
33-34: Acronyms (ARA, EPA, DPA and DHA) must be accompanied by the full name. They are only described in Table 4.
and
This also happens with other acronyms throughout the text, such as for example SFA, UFA, MUFA, PUFA
Thanks a lot, these remarks have been accepted and all the corrections have been inserted in the revised paper.
91-92: The use of this method should be supported by a bibliographical reference.
Done, two new references (i.e. ref. 12 and ref. 13) have been inserted (see lines 114-121 of the revised paper).
99-122: Review the text of the extraction procedure of fames; is it repeated two or three times. It is not clear in the text.
Accepted and amended. The extraction procedure practically consisted in 3 consecutive extraction steps. Anyway, the whole part has been carefully checked and modified in order to better explain its meaning.
173-175: Discuss and better explain the following sentence: Applying this principle, the lambs of this study should ingest 0.75 dm-3 of colostrum containing 40 g of IgG dm-3 to meet the immunity requirements.
Many thanks for this remark: the sentence has been rewritten to better clarify its meaning.
178-181: Review the explanation of FRAP/ABTS, based on inconclusive FRAP results.
Again, many thanks for this suggestion: the sentence has been completely rewritten to better clarify its meaning.
190-222: It is recommended to provide some explanations about the upper levels of Se, Zn and Cu of this study. Can they be due to fed, analysis, etc?
We acknowledge the reviewer #3 for this request. The explanations solicited have been added into the revised paper(see lines 297-310)
269: Please check the format.
Thanks. Amended.
247-248: Are these results or is it a reference?: More specifically, skimmed milk has a 6 % (ABTS) lower total antioxidant capacity than whole milk.
We agree. This sentence has been rewritten in order to better clarify its meaning (lines 281-291).
273-295: The results of the fatty acid index should be discussed (TI, AI, etc).
We agree. The discussion of the fatty acid indexes has inserted in the revised paper in lines 340-347
323-324: I think the conclusion is very pretentious that the amounts of trace elements could be an important marker of environmental contamination of an area in which animals are breed. Further analysis (soil, water, etc.) would be required to reach this conclusion.
We agree. The sentence has been deleted by the revised version of the manuscript.
Round 2
Reviewer 2 Report
Thanks to the authors for making improvements to the submitted manuscript.
In my opinion, the manuscript is acceptable for publication by Animals, MDPI.
Best wishes,
M.Wilk